# Microtia Ear Reconstruction with Patient-Specific 3D Models—A Segmentation Protocol

**DOI:** 10.3390/jcm11133591

**Published:** 2022-06-22

**Authors:** Juan Pablo Rodríguez-Arias, Alessandro Gutiérrez Venturini, Marta María Pampín Martínez, Elena Gómez García, Jesús Manuel Muñoz Caro, Maria San Basilio, Mercedes Martín Pérez, José Luis Cebrián Carretero

**Affiliations:** 1Oral and Maxillofacial Surgery Department, University Hospital La Paz, 28046 Madrid, Spain; mpampin@ucm.es (M.M.P.M.); hortensia4@hotmail.com (E.G.G.); chuslolo@yahoo.es (J.M.M.C.); mmercedes.martin@salud.madrid.org (M.M.P.); josel.cebrian@salud.madrid.org (J.L.C.C.); 2Fundación Para la Investigación Biomédica del Hospital Universitario La Paz, Calle de Pedro Rico 6, 28029 Madrid, Spain; alessa.gv@hotmail.com; 3Deparment of Pediatric Surgery, Hospital Universitario La Paz, Calle de Pedro Rico 6, 28029 Madrid, Spain; mariacarmen.sanbasilio@salud.madrid.org

**Keywords:** microtia, reconstruction, surface scan, 3D printing, surgical planning

## Abstract

(1) Background: In recent years, three-dimensional (3D) templates have replaced traditional two-dimensional (2D) templates as visual guides during intra-operative carving of the autogenous cartilage framework in microtia reconstruction. This study aims to introduce a protocol of the fabrication of patient-specific, 3D printed and sterilizable auricular models for autogenous auricular reconstruction. (2) Methods: The patient’s unaffected ear was captured with a high-resolution surface 3D scan (Artec Eva) and post-processed in order to obtain a clean surface model (STL format). In the next step, the ear was digitally mirrored, segmented and separated into its component auricle parts for reconstruction. It was disassembled into helix, antihelix, tragus and base and a physical model was 3D printed for each part. Following this segmentation, the cartilage was carved in the operating room, based on the models. (3) Results: This segmentation technique facilitates the modeling and carving of the scaffold, with adequate height, depth, width and thickness. This reduces both the surgical time and the amount of costal cartilage used. (4) Conclusions: This segmentation technique uses surface scanning and 3D printing to produce sterilizable and patient-specific 3D templates.

## 1. Introduction

Auricular malformations are severe deformities due to a deficiency of cartilage and/or skin in the ear. Microtia is the most severe form, ranging in severity from having no pinna at all to having a perfectly formed pinna but smaller than the contralateral one. Microtia incidence is estimated at 1 in 6000 live births, although the incidence varies according to ethnicity [1]. In 90% of patients, only one side is involved, with half of those appearing on the right, and in 65% of cases, microtia affects boys instead of girls [2].

Nagata proposed a microtia’s classification scheme that included lobule-type, concha-type and small concha-type terms. The lobule-type exhibits remnants of the lobule and the auricle without a canal, concha or tragus. The concha-type has a variable presence of the lobule and tragus. The small concha-type has a small indentation of the concha and the remains of the lobule and auricle [3].

Auricular reconstruction with costal cartilage has been the workhorse in ear reconstruction since the publication of Tanzer’s technique in 1959 [4]. Although Firmin and Nagata subsequently improved the technique, reducing the number of stages to two, the shape and placement of the auricular framework are still guided by the use of a template [5,6].

In 2018, Yotsuyanagi published a modification technique. He differentiated between the cartilage frame and template for lobule-type microtia and that for concha-type microtia, omitting in the latter one the lower half beneath the antihelical area and the concha cymba in the base frame while keeping the antitragus. The idea behind this is that non-deformed cartilage should be used to its fullest extent [7].

Traditionally, a 2D X-ray film template has been used to help shape the framework using the unaffected ear as reference. However, last reports have shown that the use of 3D templates facilitates and improves surgery, showing better aesthetic results and shortening surgical time [8].

Recent contributions from surface scanning and 3D printing have greatly improved the fabrication of 3D ear templates. Several groups of surgeons have described various protocols of image acquisition (CT, laser scanner, stereophotogrammetry, etc.), data processing and 3D printing of the ear template, always with the help of an engineer [9].

Surface scanning is an imaging technique that has recently burst into healthcare, which computes the spatial coordinates of the patient’s anatomical surface in order to create a 3D digital model. The main advantage of this technique is the non-radiation, the high accuracy and resolution and the short acquisition time. Moreover, 3D printing allows the creation of a physical model of said scanned surfaces, which is useful for surgical planning.

The aim of the present study is to describe a protocol to fabricate patient-specific models and segmented templates, based on the opposite unaffected ear, using a surface scanner, computer-aided design (CAD) software and 3D printing.

We also present two representative clinical cases of each type of microtia from the classical Nagata classification, to correlate it with the two types of technical modifications for digital modeling of the template, based on Yotsuyanagi’s reconstruction technique modifications.

## 2. Materials and Methods

In this section, the proposed protocol is explained, divided into four different stages: surface scanning, models preparation, 3D printing and surgery.

### 2.1. Surface Scanning of Auricles

The first step of the protocol consists in the image acquisition of the healthy ear of the subject. High accuracy is needed to capture the detailed features of the auricular structure, which are essential for a successful reconstruction.

Specifically, a structured-light surface 3D scanner (Artec Eva, Artec Group, Luxembourg) was used to obtain the auricular structure of the microtia and normal auricles. This scanner allows the capturing of surfaces with a 3D resolution up to 0.2 mm, at a 16 fps reconstruction rate and also saving the model’s texture. Our average image acquisition time was 65 s.

The obtained point cloud data were processed in the native Artec Studio 16 Professional software (Artec Group, Luxembourg), following different steps. The erasing tool allows the user to manually discard the undesired areas of the point cloud data, like the background. Then the registration tool analyses all the frames captured during the scanning process and aligns them. When the outlier removal tool is used, the noise points around the subject are removed. Next, a fusion of all the frames into a single mesh is performed. Finally, texture from the initial scan is applied to the mesh. Figure 1 shows the original patient’s scan and the final textured mesh.

For the next stage, the obtained mesh was exported as a STL file.

### 2.2. Digital Models Preparation

The exported mesh was then processed using the modeling software Meshmixer (v 3.5, Autodesk Inc., San Rafael, CA, USA).

First, the ear was refined, removing the noise and filling the defective areas in order to obtain a closed and clean mesh. Then this mesh, corresponding to the healthy ear, was mirrored to design the new components of the patient’s affected ear. These components are the anatomical parts considered essential for surgical planning: helix, anti-helix (with both superior and inferior crura) and tragus. Once these parts were segmented and sculpted in detail, two small notches were added to guide the positioning of the helix and anti-helix. Neither the navicular fossa nor the scaphoid fossa were carved.

Moreover, a sheet or base frame model was also extracted from the ear model, by extruding (1.2 mm thick) the intersection between the ear and a transverse plane.

Finally, these 5 models (ear, helix, anti-helix, tragus and base frame, shown in Figure 2) were exported in STL format for the 3D printing stage.

We have compiled a repository of images and 3D models in STL format of previous bilateral microtia patients, from which we can access and select the model that best suits each patient, according to age, height, stature and ethnicity.

### 2.3. 3D Printing of the Models

The digital models were printed in a stereolithography (SLA) 3D printer (Form 2, Formlabs, Somerville, MA, USA) with Surgical Guide resin (Formlabs) at a 0.1 mm printing resolution (Figure 3).

After printing, the models were removed from the build platform and washed for 20 min in a Form Wash (Formlabs) filled with 99% isopropyl alcohol, to clean the parts and remove the liquid resin. Then they were post-cured at 60 °C for 30 min in a Form Cure (Formlabs) to achieve biocompatibility and optimal mechanical properties.

Finally, the models were sterilized and used in the operating room.

### 2.4. Framework Fabrication

The size of the normal auricle and the type of microtia determined how many cartilages were harvested. The 6th, 7th, 8th and 9th costal cartilages are the most commonly used.

The framework consisted of four main parts: the base frame, tragus, helix and antihelix. The 6th costal cartilage was used to carve the antihelix, while 7th and 8th were used for the base frame and the tragus. Lastly, the helix was made from the 9th costal cartilage (the longest one) (Figure 4a). Thanks to these 3D models, several cartilage fragments were left over and repositioned in the rib pocket so that they could be used as Firmin’s P1 in the second stage (Figure 4b).

## 3. Results

The sterilized, patient-specific models were placed next to the cartilage grafts, where the surgeon could hold them, rotate them and analyze their shapes and relief. Although it was not considered and compared in this paper, we believe the preoperative planning of the surgery during the joint digital processing with the engineer allowed the saving of both surgical time and the amount of cartilage, without losing the artistic element inherent in this type of surgery.

### 3.1. Case Report: Case 1

The patient was an 8-year-old male with lobule-type microtia on his right side (Figure 5). The models were prepared using the method described in point 2.

The 6th, 7th, 8th and 9th costal cartilages from the right side were harvested and, following the shape, depth and 3D curvatures of the models, the four anatomical parts of the ear were carved independently. The helix part was attached to the tragus (Figure 6a), and both were fixed to the base frame using stainless steel wire. The antihelix was joined in the same way (Figure 6b).

The reconstructed pinna appeared harmonious with a satisfactory shape 20 days after surgery (Figure 7). As it can be appreciated, residual tissue was limited, and the earlobe remained unnatural in shape. A small transposition during the second stage will probably be needed.

### 3.2. Case Reports: Case 2

The patient was a 7-year-old male with concha-type microtia on his right side (Figure 8a).

First, the double V–Y advancement flap technique was performed to extend the skin cover of the frame, as described by Duan in 2019, initially for the correction of Tanzer type IIB constricted ears. (Figure 8b) [10].

The models were prepared using the method described in Section 2 (Figure 9a). However, following Yotsuyanagi’s technique, a base frame was designed and carved, in which the antihelix area and concha were omitted (Figure 9b). The residual tissue and remnant cartilage were utilized.

Subsequently, the framework was covered using the temporoparietal fascia and a scalp skin graft. The skin cover was molded back to the structure using suction probes (Figure 10).

## 4. Discussion

For a head and neck reconstructive surgeon, auricular reconstruction in patients with microtia is one of the most technically challenging and demanding procedures. It requires a steep learning curve, due to the extremely complex and unique geometry of the procedure, which strongly depends on the artistry and technical skills of the surgeon.

One of the main difficulties is to transfer the 3D characteristics of the anatomical elements of the ear (thickness, height, depth, reliefs) from a 2D template to the cartilaginous framework.

To overcome these limitations, Kelley first developed a 3D template for pinna reconstruction in 1998. Thereafter, studies have focused on scanning the contralateral healthy ear and fabricating a patient-specific 3D model that can guide the reconstruction procedure and ensure satisfactory aesthetic results while reducing surgical time and procedural complexity [11].

Several scanning techniques have been described, such as diagnostic imaging techniques (CT or MRI) [12], 3D surface laser scanning [13] and photogrammetry techniques [14]. CT has some disadvantages, such as its high cost or the patient´s exposure to radiation. Considering that most of the patients undergoing ear reconstruction are infants and adolescents, there is concern about the effects of radiation exposure, such as an increased risk of tumors of the central nervous system, leukemia and lymphoma, although this is still unclear [15].

Consequently, considering the complex structures and intricate details of the pinna, a suitable method for acquiring the shape of the outer ear is to use photogrammetry techniques, providing an indicated point accuracy ranging between 0.05 and 1.0 mm and a resolution between 27.9 and 68.3 polygons/mm^2^, depending on the device [16]. The Artec device was used in our study. This system provides high precision and accuracy in ear scanning. However, a study comparing various systems found that cheaper and more accessible systems (such as the iPhone) showed impressive results, considering it as an alternative to more expensive systems [16]. There have been subsequent publications that confirm the exactness and reliability of the systems, with similar acquisition times, making these systems more accessible and convenient for everyday practice [17].

It is true that this is a reference system, with high accuracy and a short acquisition time (about 60 s on average). Despite the fact that our patients are around 8 years old and are cooperative enough to remain still for a minute or a minute and a half, there are other technologies, such as three-dimensional (3D) stereophotogrammetry, which is both faster and noninvasive and has many advantages, such as its speed of capture and ability to produce high-resolution photorealistic images. The system is composed of five cameras positioned at different angles to the subject. The images are captured simultaneously by these two-dimensional (2D) digital cameras and the 3D result is displayed. It may be an alternative to consider in patients who are not very cooperative [18].

As alternatives, Medpor (porous polyethylene material) reconstruction can be utilized, which can offer high-quality reconstruction with even better aesthetic results. However, it is still less widely used due to concerns about device exposure and possible morbidity; therefore, many surgeons choose not to use porous polyethylene. This can severely limit the options available for reconstruction following salvage surgery [19].

Another reconstructive technique is tissue-engineered reconstruction. Although promising, it is true that it is still a long way off and further improvements have to be made. Almost 20 years ago, a picture of a rat with an ear on its side captured the imagination of the world [20]. However, a solution to its main negative impact (resorption of the cartilage) has remained elusive. Its unsatisfactory clinical efficacy is due to reconstruction constructs easily causing inflammation and deformation. As one of the last approaches to solving this issue, Jia et al., used auricular chondrocytes in conjunction with a bioactive bioink based on a biomimetic microporous methacrylate modified acellular cartilage matrix (ACMMA) to produce biological auricle equivalents with precise shapes and low immunogenicity [21]. To date, only six patients have been published with good aesthetic results and without resorption [22].

Our protocol allows predictable ear reconstruction and the optimization of cartilage harvest, offering satisfactory aesthetic results. Tree-dimensional templates have shown to reduce surgery time, meaning savings in hospital resources, which could offset the main cost of the printer and the scanner type chosen. Furthermore, three-dimensional modeling enables computer programs to quickly extrapolate missing information, eliminate surrounding tissues and design and fix prosthetics accurately.

A key step in this technique is the choice of the type of reconstruction and, therefore, of the template, depending on the type of auricular anomalia. Yotsuyanagi´s modification technique facilitates the analysis, the selection of the reconstruction framework and its milling. In addition, it favors the protocolization of the digital phase, thereby standardizing reconstruction. Depending on the type of microtia, segmentation also differs, requiring a learning curve for the correct fabrication of the PS model, allowing it to project aesthetically under the skin.

In Case 2, it was decided not to add the antihelix to the frame since the native antihelix structure was present. Nonetheless, it can be discussed that a more marked effect could have been achieved through an antihelix reconstruction and remanent cartilage removal.

In cases of bilateral microtia, we are developing a repository of images and 3D models in STL format of our previously treated patients. This would allow patients to access these models and select the one that is most suitable for each patient, based on age, height, stature and ethnicity. In the future, we would like to be able to share this repository as a library, so that any team could access and share their cases or look for what might be useful to their patients.

However, it must be considered that this protocol is technically demanding, requiring knowledge in 3D design and therefore also has a considerable learning curve, especially during the preoperative planning stage when the engineer and the surgeon work together on the digital segmentation of the ear. Identifying the prerequisites for a successful simulation of ear reconstruction, Mussi et al., have designed a strategy for an interactive design and customization of this procedure with the objective of semi-automation, in this case, by developing patient-specific reconstruction trainers [23].

Furthermore, it would be necessary to analyze what depth and thickness should be applied to the helix, antihelix, base, etc., and, depending on this, what esthetic result we would obtain in the projection once it is covered by the skin. There are groups working on this and it will be something we will take into account in future interventions [24,25].

As improvements to this study, it would be interesting to increase the number of cases and measure and analyze surgical time, since 3D templates have been shown to reduce surgery time, resulting in a lower risk of complications and shorter hospital stays [8,26,27]. In order to quantify better surgical outcomes, it would be interesting to develop a prospective study that compares these results with the conventional method, which calculates the mean distance between two points.

## 5. Conclusions

Our study presents the experience of a tertiary care center that used this segmentation technique to produce sterilizable, patient-specific 3D ears using surface scanning and 3D printing.

The study of a larger sample may permit a more thorough understanding of whether patient-specific 3D models reduce operating times.

Surgeons, hospitals and legislators should aim to reduce operating times as a universal goal, not only to improve patient outcomes by reducing complications but also to reduce costs.

## Figures and Tables

**Figure 1 jcm-11-03591-f001:**
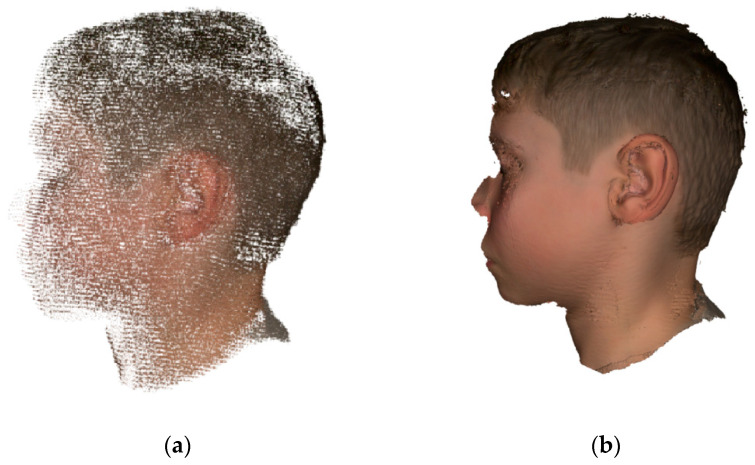
Surface scanning of the healthy ear of the subject with the Artec Studio software: (**a**) first captured point cloud data before post-processing; (**b**) final reconstruction of the 3D digital model after post-processing.

**Figure 2 jcm-11-03591-f002:**
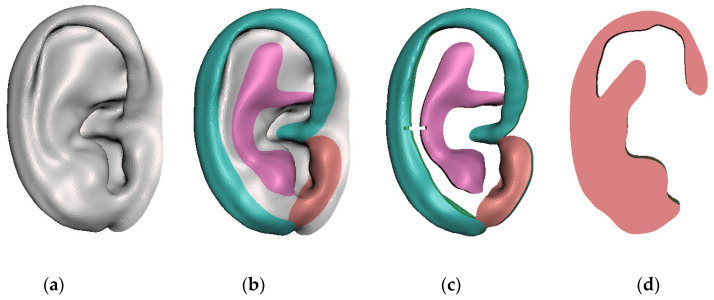
Modelling of the ear and the anatomical parts for the surgical planning: (**a**) refined and mirrored healthy ear; (**b**) segmentation of the helix (blue), anti-helix (purple) and tragus (brown); (**c**) sculpted anatomical parts; (**d**) base frame model.

**Figure 3 jcm-11-03591-f003:**
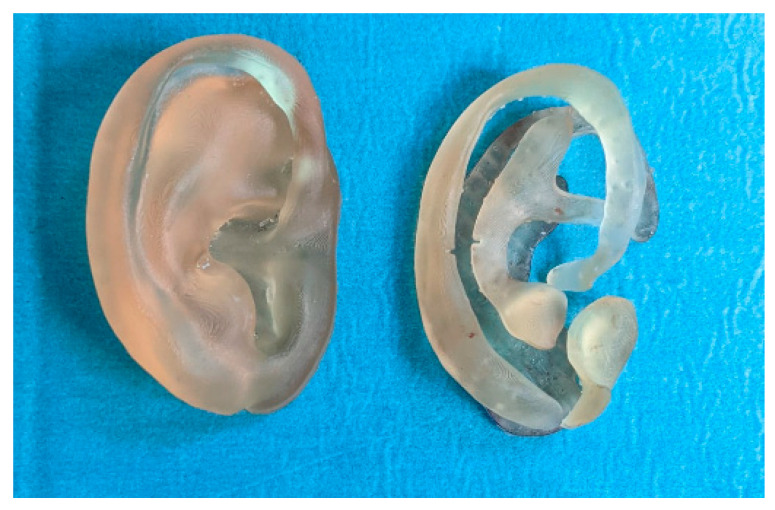
3D printed models of the ear (**left**) and the different anatomical parts (**right**).

**Figure 4 jcm-11-03591-f004:**
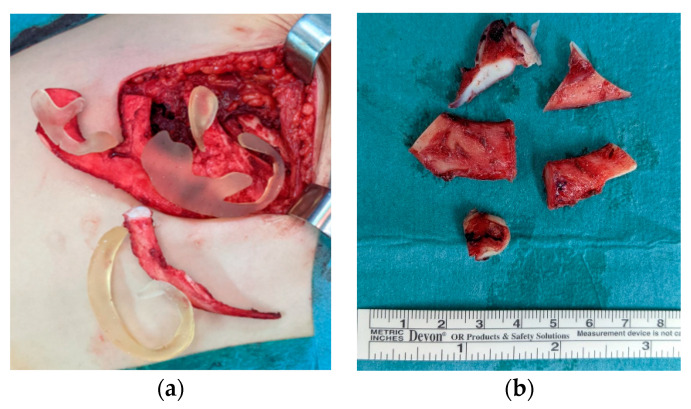
(**a**) 3D model over the costal cartilage during the harvesting; (**b**) cartilage excess.

**Figure 5 jcm-11-03591-f005:**
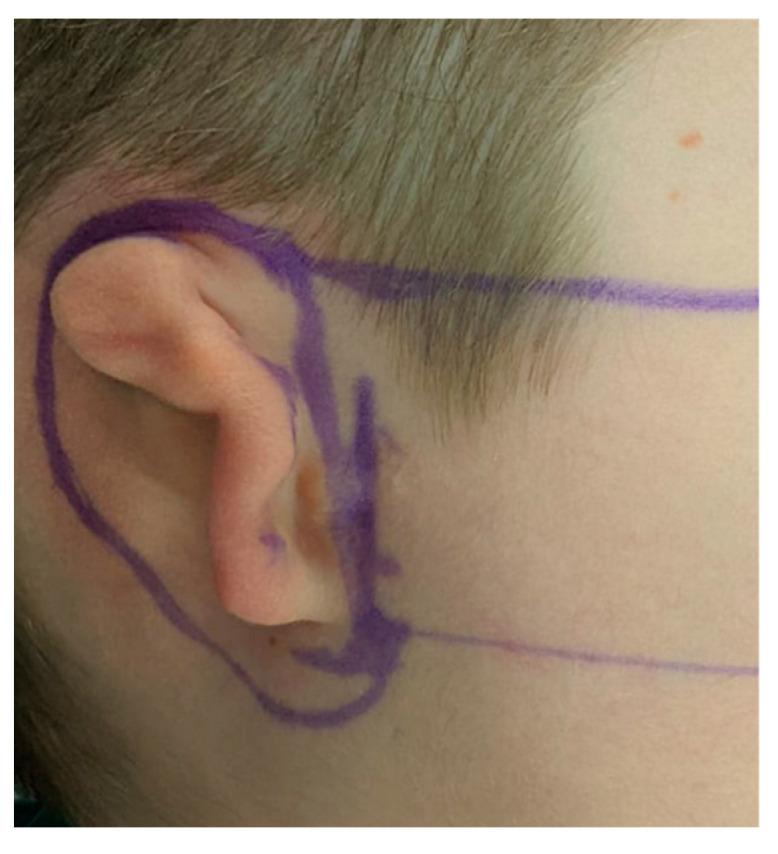
Case 1. Preoperative oblique view of lobule-type microtia.

**Figure 6 jcm-11-03591-f006:**
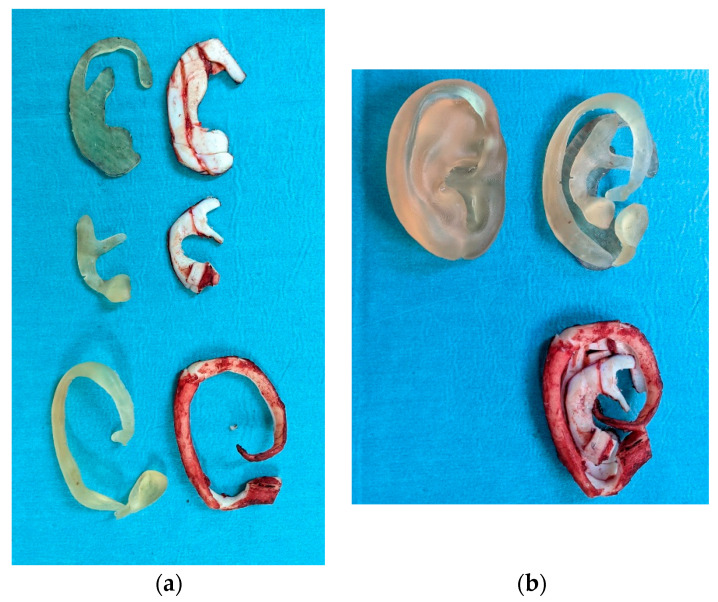
Case 1: (**a**) Each patient-specific 3D model together with its cartilaginous counterpart.; (**b**) final framework, with the two segmented and non-segmented models.

**Figure 7 jcm-11-03591-f007:**
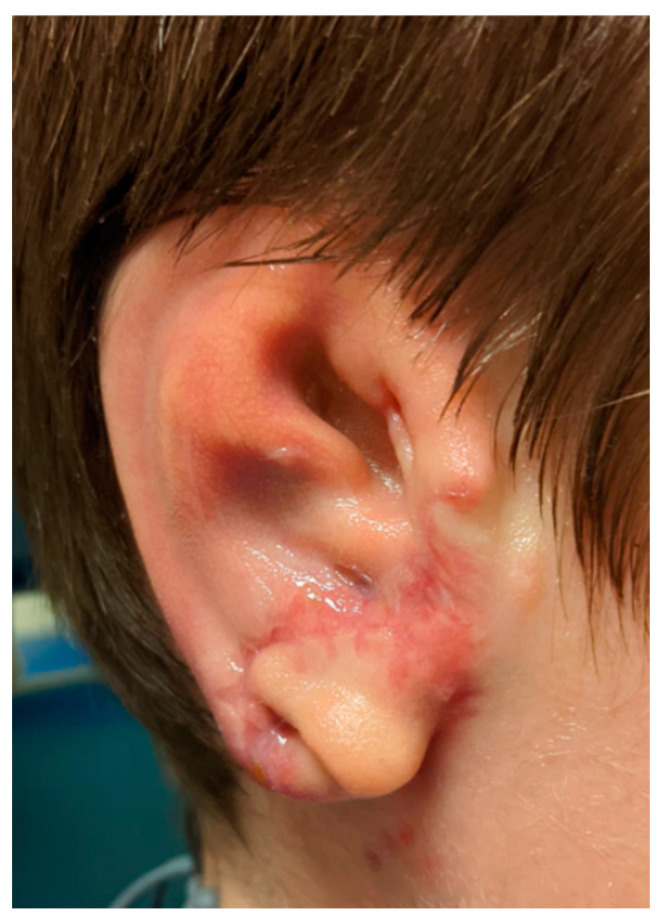
Case 1: Postoperative oblique view, 20 days after the first stage.

**Figure 8 jcm-11-03591-f008:**
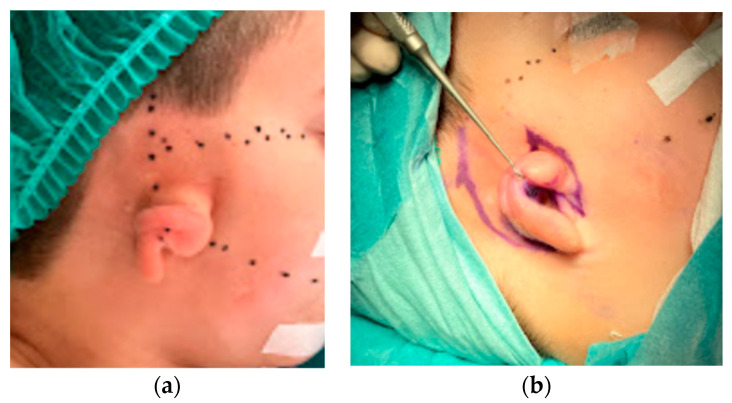
Case 2: (**a**) Preoperative lateral view of concha-type microtia; (**b**) design of the skin incision.

**Figure 9 jcm-11-03591-f009:**
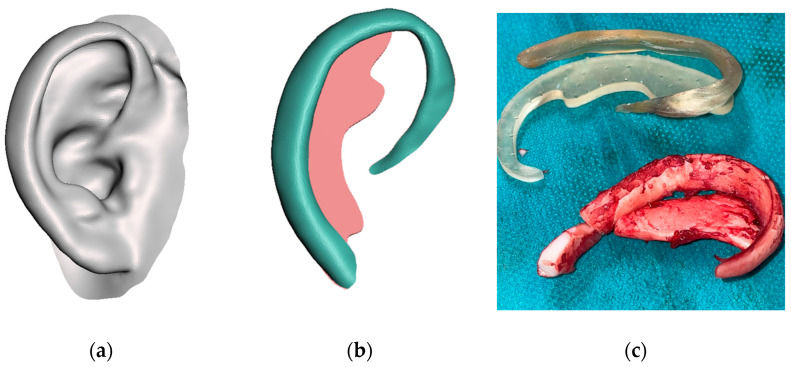
(**a**) Refined and mirrored healthy ear; (**b**) patient-specific base frame and helix models; (**c**) costal cartilage framework.

**Figure 10 jcm-11-03591-f010:**
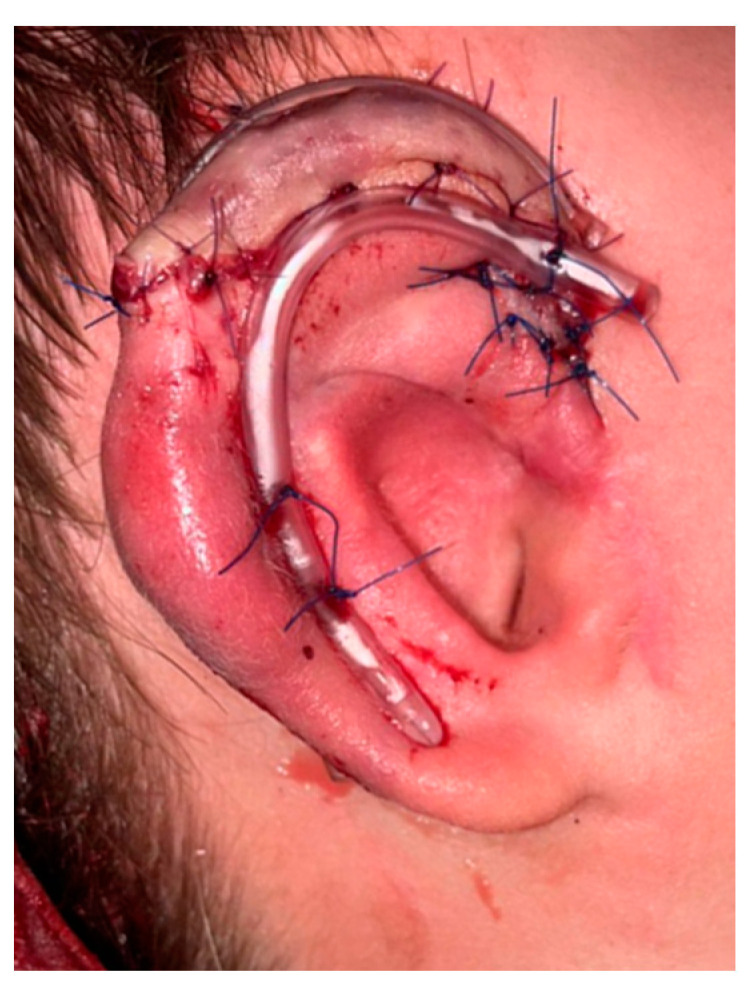
Postoperative oblique view, 10 days after surgery.

## Data Availability

Data is contained within the article.

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
