# Peer review of "Microtia Ear Reconstruction with Patient-Specific 3D Models—A Segmentation Protocol"

_jcm, 2022, doi:10.3390/jcm11133591_

Round 1
Reviewer 1 Report
The article title "Microtia Ear Reconstruction with Patient-Specific 3D models. A Segmentation Protocol" is a patient specific research done for defective ears or microtia ear. The paper is well structured and scientifically sound. The introduction, results and discussion are well structured and interesting to the readers. The authors can include few details from the following recently published articles related to their topic.
1. You P, Liu YC, Silva RC. Fabrication of 3D models for microtia reconstruction using smartphone-based technology. Annals of Otology, Rhinology & Laryngology. 2022 Apr;131(4):373-8.
2. Kraai T, Vandenberg K, Lewin S, Seelaus R. Microtia Reconstruction in Patients with Craniofacial Microsomia. InCraniofacial Microsomia and Treacher Collins Syndrome 2022 (pp. 177-236). Springer, Cham.
3. Mussi E, Servi M, Facchini F, Volpe Y, Furferi R. A rapid prototyping approach for custom training of autologous ear reconstruction. International Journal on Interactive Design and Manufacturing (IJIDeM). 2021 Dec;15(4):577-85.
Author Response
Good afternoon,
first of all, thank you very much for reading and for the suggestions offered.
I have added some details, mentioning two of your input on lines 275 and 328.
Kind regards

Reviewer 2 Report
In the present study, the authors present a protocol of fabrication of patient-specific, 3D printed and sterilizable auricular models for autogenous auricular reconstruction.
They proposed the concept of ear reconstruction with a high-resolution surface 3D scan (Artec Eva) to segment into its components and print a physical 3D model of each part. Following this segmentation, the models were used as guides in the operating room. Therefore, they expressed a considerable helping potential for reconstructions operating and other surgical procedures using 3D guides.
In my opinion, they also should consider a situation, when patients need to reconstruct both ears. What should be modified in the proposed protocol to use it for the operation of reconstructing the patient's both ears? Maybe in such a situation, this protocol would be useful too.
The paper is very well prepared, but some issues should be improved. For example, the legends of Figures 2, 7 and 9 should be on the same page. Also, the Authors should unify the writing reference in the literature. It is written in the square bracket sometimes after a dot (e.g. line 270 or 274), but sometimes it is before a dot at the end of sentences (e.g. line 286). I suggest putting it before the dot at the end of sentences.
Author Response
Good afternoon,
first of all, thank you very much for your attention and comments.
All your suggestions have been added
Line 122 and 313
In cases of bilateral microtia, we are developing a repository of images and 3D models in STL format of our previously treated patients. This would allow patients to access these models and select the one that is most suitable for each patient, based on age, height, stature, and ethnicity. In the future, we would like to be able to share this repository, as a library, so that any team could access and share their cases or look for what might be useful to their patients.

Reviewer 3 Report
I have read with attention and interest this manuscript which focuses on a 3D protocol for the reconstruction and surgery of ears. I found the article interesting and well written.
However, I have some comments and questions for the authors that might help them to add important information and which are reported below.
Major comments:
1) There must be some details I lost during the read: the protocol is well described but if I have understood well the protocol was applied only to two real cases? In this case this should be highlighted better when the authors present the aim of the study and also explain clearly in M&M.
2) I can see two surgical cases where the protocol have been applied:
Have the authors compared results among the post-operative 3D model of the ear underwent to surgical treatment with the healt contro-lateral one to quantify the surgical outcomes? using the superimposition protocol they could calculate the Root Mean Square among the two ears (the average point-to point distance) for quantifing better the surgical succes?
Minor comments:
1) Row 58: I suggest to add to CT and laser scanner also stereophotogrammetry.
2) How long does it take to capture the 3D ear surface? I mean, can the device acquire the 3D model in few seconds, or more time is required? This information should be added to M&M and also added in discussions. In fact, if the acquisition is instantaneous (as for stereophotogrammetry) that the author could describe this fact as an additional advantage (in addition to the safety and accuracy of the methodology). On the contrary, if the process takes more time (the acquisition is not immediate), then this could affect the final model because motion by scanned subjects can occur, especially if the patients are not collaborative (like children or special need patients) and this point should be seen as possible limitation and so added to discusisions.
However, in both cases, even if oppost, the authors could add in discussions a brief comment on this important point.
Author Response
Good afternoon,
first of all, thank you very much for your attention and comments.
Major comments:
- This protocol with these two types of modification we have done them in few cases and we present these two as the most paradigmatic for each type of reconstruction.
- Indeed, the idea of comparing the pre and postoperative result would be very illustrative. As we have few cases performed in this way, the idea is to accumulate more cases and perform the comparison in several cases and calculate the Root Mean Square once the second time has been performed in all of them.
We could even compare it with other cases performed in the traditional way. We are still thinking about the design of the study. I have tried to check it on the line 315.
Minor comments:
- Added, thank you.
- Added, line 272
